# Impacts of community forestry on forest condition: Evidence from Sri Lanka's intermediate zone

**E. M. B. P. Ekanayake[1,2], G. T. Cirella[3], Yi Xie[1]** *

**1** School of Economics and Management, Beijing Forestry University, Beijing, China, **2** Department of Forest Conservation, Sampathpaya, Battaramulla, Sri Lanka, **3** Faculty of Economics, University of Gdansk, Sopot, Poland

☯ These authors contributed equally to this work.

* yixie@bjfu.edu.cn

**Data Availability Statement:** All relevant data are within the manuscript and its Supporting Information files.

**Funding:** Xie. This work was supported by the Soft Science Funds from National Forestry and

## Abstract

Sri Lanka's community forestry (CF) program emerged in the early 1980s following a global trend to conserve forest resources and provide benefits to the local community. However, very little is known about the effect of CF on forest resources. We assess the impacts of CF on forest conditions of semi-mixed evergreen forest in the intermediate zone of Sri Lanka using the before-after control-impact method. The study examines tree density, regeneration, woody species diversity, and evidence of disturbance as parameters to analyze the impact of the CF program. Data are analyzed using the difference in differences approach. The results show that the CF program has increased seedling and sapling density to a significant degree and reduced human disturbances. A major contribution of the CF program is that it was found to reduce invasive species and forest fires. The program reduced the amount of invasive species up to six times less than previous. The findings revealed that the impact of CF on forests may vary depending on pre-existing forest conditions, length of period to implement, perception, and decisions by local people. Community understanding and decision-making, in tandem with government policy, will weigh heavily on its future effectiveness.

## Introduction

In the late 1970s, the concept of social forestry rose as a new paradigm for forest management [1, 2]. The term was used to reflect concerns that forestry should pay closer attention to the socioeconomic welfare of rural communities [2]. Several years later, social forestry became less common and other terms more fashionable, such as community forestry (CF) and participatory forestry management (PFM). The current concept of CF has improving ecological sustainability and increasing local people's benefits as central goals, which are achieved by granting communities some degree of formal responsibility and authority for forest management [3].

Grassland Administration [grant number 2019131025], www.forestry.gov.cn. The funders had no role in study design, data collection and analysis, decision to publish, or preparation of the manuscript.

**Competing interests:** The authors have declared that no competing interests exist.

Initially, CF-oriented programs sprang up from approaches that stimulated forestry development throughout tropical and subtropical regions [4]. Seen as a strategy for stimulating rural development, alleviating rural poverty, increasing social justice, empowering women, and sustainable forest management, the concept was gradually incorporated worldwide [5–7]. At present, considerable amounts of forest lands are designated under CF. For example, throughout the Asia Pacific, 25% of forest lands are managed by communities and indigenous people [8]. As one of a handful of countries to apply this concept, Sri Lanka in the latter part of the 1980s amended and revised its supportive policies by opening up provision for community involvement in forest management [9]. Since 2003, the Department of Forest Conservation (FD), a non-ministerial government department responsible for forestry in Sri Lanka, has been testing and trialing various PFM approaches using the CF model—including the Sri Lanka Australian Natural Resource Management Project (SLANRMP) implemented from 2003 to 2009. SLANRMP's aim was to improve forest management by way of local community support and poverty alleviation of rural people living in dry and intermediary climatic zones. According to its activity report, from 2008, SLANRMP made substantial contribution to the livelihood of local communities and management of forest resources. To this effect, in 2008 the FD decided to expand the CF program with its own funding. Currently, the program has been implemented in 18 districts across 167 CF sites, benefiting approximately 125,000 individuals. Nearly 23,500 ha of forest lands are managed by the CF program as a buffer zone for planting, farm wood lots, and enrichment planting [10]. As such, there is increasing evidence that CF has, in many cases, been more successful in forest conservation and community development than centralized, state-driven management [11–13]. In fact, control and restriction-based policies used by state-run forests have negatively affected forest dwellers, in terms of their economy and livelihood, and in a number of occasions resulted in the overexploitation of forests and forest resources [14–16].

Globally, a growing number of scholars affirm several reasons why CF, in conjunction with forest management, functions well [17, 18]. For instance, community management, generally, costs less and requires less resources to protect and conserve forests than state-level management [19]. The CF program has contributed to the livelihood of local people by providing subsistence needs and offering a possible pathway to eliminating poverty [20–22]. Other studies suggest that participation of local communities in forest management enhances forest conditions due to the fact that their indigenous knowledge, especially with regard to the environment, can assist in developing and implementing proactive management strategies [23–25]. Also, their ability to access forests provides them with a relatively easy way to monitor and target illegal and unsustainable forest and resource use [17]. Moreover, in several cultures such as Hinduism and Buddhism forests are consciously protected as sacred, which denotes those communities' vested interest in their conservation as well as any associated forest repository [25, 26]. However, evidence for the impact of CF on local socioeconomics and the environment is mixed, with studies reporting both positive and negative impacts [27, 28]. Furthermore, research on impact evaluation of CF has largely addressed socioeconomic influences as compared to environment-oriented ones [29, 30]. A key reason for this discrepancy is it is much more challenging to quantify the impacts on the environment [31]. Based on a review of the literature, CF has been found to be successful in improving forest and forest resources. Studies have reported that CF programs have the potential of improving and conserving the forest ecosystem [32–35]. Dougill et al. [36] reported that CF approaches could improve the forest ecosystem in terms of regeneration. Similarly, Gobeze et al. [37] found that CF effectively reduces forest degradation. There is also evidence for the success of CF in biodiversity conservation [38–40] and forest carbon management [14, 41, 42], while Vianna and Fearnside [43] argued it has low impact on biomass carbon stock of managed vegetation. Also, Gatiso

[17] asserted that the heavy dependence of CF members on forest products may undermine the success of the CF program by leading to forest resource degradation. However, Bowler et al. [30] reported that evidence-based conservation under CF still remains relatively poor.

Development of the CF program in Sri Lanka has closely examined the socioeconomic effectiveness of community-based forest management at the individual and community level [44]. Several field studies have reported that the livelihood benefits provided by CF were conducive to enhancing the living standard of rural communities [45, 46], optimizing local resource use, and contributing to financial community-oriented transactions [44]. On the other hand, some findings indicate that uneven distribution of economic benefits among different groups has satisfied the interests of a very few while marginalizing the majority—often contributing to conflict [46, 47]. Within Sri Lanka, a very limited number of studies have emphasized the evaluation of impacts of the CF program on the environment [44].

The focus of this paper examines semi-mixed evergreen forest, where the majority of CF sites are established. Even though this forest type covers four-fifths of Sri Lanka's vegetation, it has been less intensively studied than the vegetation comprising the other 20% [48, 49]. Globally, semi-mixed evergreen forests are distributed in the tropics and subtropics. Impacts by human disturbance, uncontrolled shifting cultivation, and illegal logging are higher in semi-mixed evergreen forests than in evergreen forests [50–52]. Despite tropical and subtropical forests attracting attention from a large number of researchers, studies on the semi-mixed evergreen forest are comparatively limited [53]. In Sri Lanka, 40% of the rural population live throughout the dry zone and intermediate zone (IZ) of the country, largely dependent on semi-mixed evergreen forests for wood, non-timber forest products (NTFPs), and fodder [54, 55]. A study by Ekanayake et al. [16] revealed that semi-mixed evergreen forest contributes to 11.1% of the total income of rural households; however, very little is known about the impact of CF in these forest lands. Findings are hindered by insufficient baseline data (i.e., before the program started) on the forest conditions [10]. It is evident that due to limited data, the effectiveness of the CF program in improving the forest ecosystem cannot be guaranteed [56–58]. Hence, this study investigates the impact of the recent CF program on the dynamics of forest conditions, through a comparative study of controlled and experimental design, applied in nine different CF sites in the IZ of Sri Lanka.

An impact analysis of CF on the forest condition is novel in two ways. First, it provides insight into CF impacts (i.e., where the majority of CF sites are established) on floristic composition and diversity of semi-mixed evergreen forest. This complements much of the previous work on impact of CF on the environment and forest condition via community perception—especially research focused on ecosystem services and forest product provision [44, 46]. Second, a comparative approach is conducted using the before-after control-impact (BACI) method. As such, globally very few quantitative studies are available [20, 30, 34] and none are reported in Sri Lanka. These gaps in the research warrant examination by building upon the state of the art and presenting best practices research. A breakdown of the paper is structured as follows: section 2 contains the methodology, section 3 illustrates the results, section 4 discusses the findings and compares them with empirical literature, and section 5 the conclusion.

## Methodology

### Conceptual framework

Forests are increasingly viewed as vital capital assets that provide a wide range of ecosystem services valued by people [59]. The condition of forest systems is valued by people, mostly due to the fact that they can help sustain and protect human life as well as improve quality of life. Forest condition derives from the structure and functionality of the forest systems (e.g., tree

abundance and rates of transpiration), outputs that people desire (e.g., non-timber forest products (NTFPs)), and people's impact on it (e.g., felling trees) [60, 61]. To assess the effects of CF on forest condition a developed conceptual framework based upon the accountability of decentralization by Agrawal and Ribot [62] and Schlager and Ostrom [63] is proposed. The conceptual framework draws on insights from literature review and previous studies directly concerned with the impact of CF on forest condition (Fig 1).

Before implementation of the CF program, all forest management activities such as reforestation, forest fire management, controlling forest offences, boundary demarcation, regulation of forest product extraction, etc. are done according to the control and restriction-based policies and command-based decisions of FD. The implementation of the CF program transfers power to make decisions about forest resources downwardly to accountable local authorities—namely community-based organizations. Forests in Sri Lanka where the CF program is implemented are controlled by their communities in most respects, freeing them from the command-based decisions of FD that are typical for state-managed forests. Due to CF, power to apply government policy and implement rules to ensure compliance is also handed down to community members. As a result, community members gain rights of access to forest resources, to manage them, and to have the ability to forego resource withdrawal and

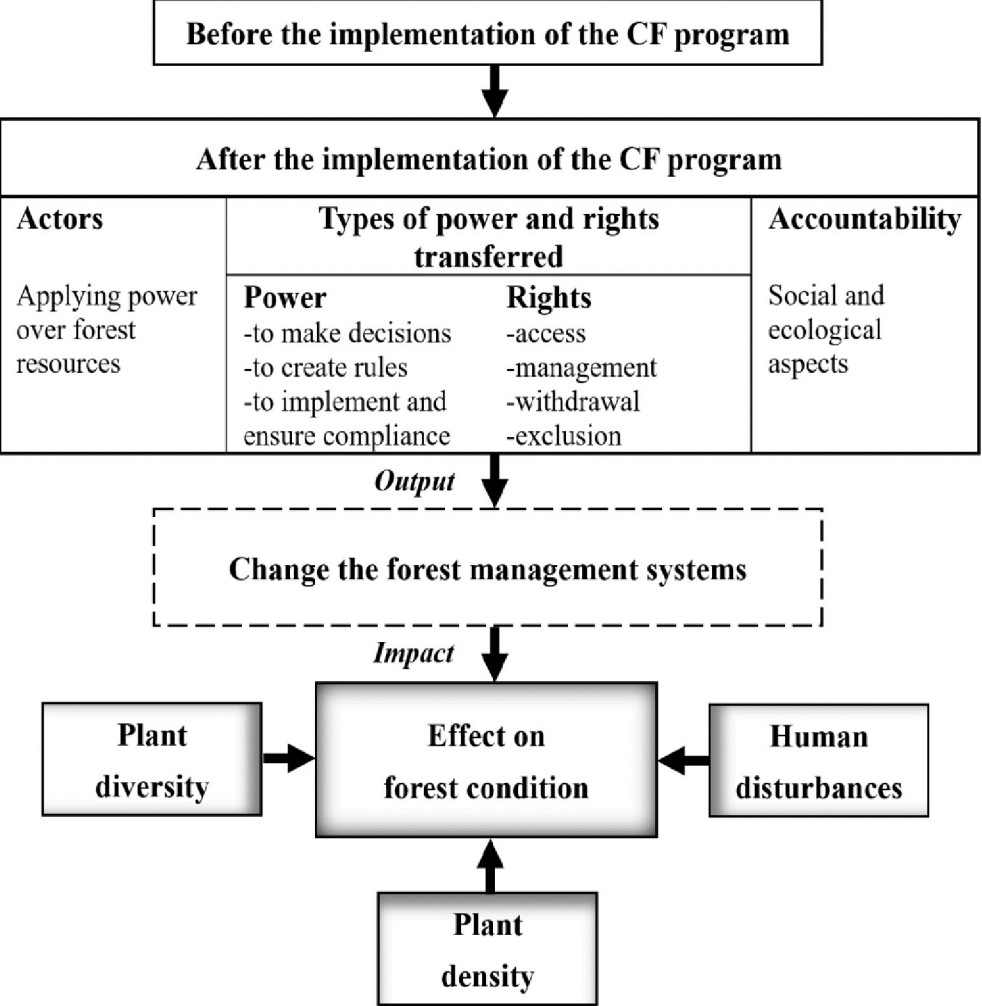

**Fig 1. Conceptual framework, adapted from Agrawal and Ribot [62] and Schlager and Ostrom [63].**

exclusion. As shown in Fig 1, changes in power and rights result in changes in the forest management system which, in turn, can be measured by social and ecological characteristics [64, 65]. As such, alteration of management systems directly influences forest condition [34, 66] and assessing existing ecological parameters is one of the well-known methods to investigate the impact of different management systems on forest condition [67, 68]. Parameters are tools which can be applied to gather and organize information in a manner that makes it useful in conceptualizing, evaluating, and implementing best forest practices [69, 70]. Parameters, especially plant diversity and composition, are retained as most valuable in forest condition analysis due to their involvement (i.e., in ecological structures, functions, and processes) as well as their significance for the forest ecosystem [71, 72]. We incorporate these parameters into the model to identify impacts of the CF program on forest condition. Moreover, several studies that applied indicators of human disturbances to measure the condition of the forest are considered [34, 73, 74].

The Sri Lanka Community Forestry Program (SLCFP) commenced in 2012 and was completed in December 2016. The SLCFP was an evolutionary step, after the SLANRMP, with specific objectives of improving the management of natural resources, supporting community livelihoods, and contributing to poverty reduction. The management of the SLCFP was contracted to the United Nations Development Programme and implemented by the FD [75]. After the identification of suitable sites, CF management plans were prepared and implemented by community-based organization members—including those involved in forest management and community development activities. The program provided support to implement livelihood-oriented development activities, improve infrastructure, and implement sound forestry activities. In each activity labor contribution was paid and material and equipment supplied [76]. The forestry component included planting of buffer zones, enrichments, firebreaks, live fences, and farm woodlots. Where farm woodlots were involved, a lease agreement was signed with the FD to ensure a thirty-year leasehold for tree tenure rights was offered to farmers for pruning branches and thinning operations of trees normalized to 80% of tree maturity [75]. In Sri Lanka, forests entered into the CF program were previously state-managed forests. Under state management, the majority of natural forests were protected, and forestry operations such as extraction (e.g. timber) were restricted. In addition, boundary demarcation and fire-belt establishment were conducted by FD using forest laborers. However, due to the high labor cost and limited human and other resources, FD is unable to maintain fire belts throughout the year. To overcome these issues, FD encourages CF members to maintain the fire belts in their CF blocks. Notably, forest blocks which were handed over to the communities were mostly degraded lands and were closer, on average, to the communities than the state-managed blocks. The FD aims to improve these degraded lands through community participation. It should also be noted that when the CF program was implemented in forest blocks, the adjacent state-managed forest blocks remained subject to the same FD management activities as before the CF program.

## Geography and study area

Sri Lanka is divided into three main climatic zones (i.e., dry, wet, and intermediate) based on seasonal rainfall. The IZ, where our study was conducted, is sandwiched between the wet and dry zones (Fig 2). The mean annual rainfall in the IZ is between 1,750 and 2,500 mm with a short and less prominent dry period [77]. The IZ has an average temperature of 30°C, ranging from 28°C to 32°C, with the highest temperatures measured in July and August. Due to topographic variation, the IZ is divided into three regions based on elevation: low country (i.e., 0–300 m), mid-country (i.e., 300–900 m), and upcountry (i.e., over 900 m). Reddish-brown

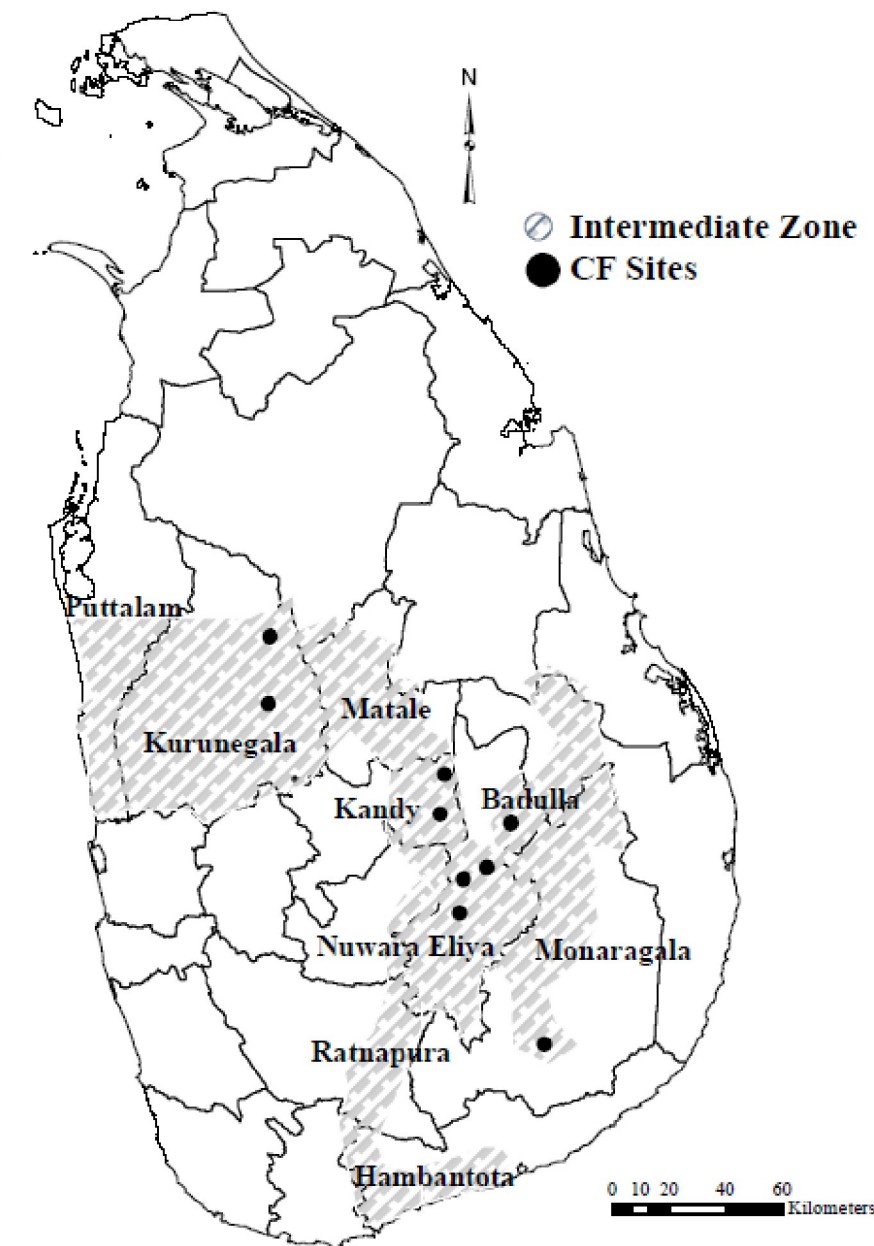

**Fig 2. Sketched map of the IZ in Sri Lanka.**

earth and reddish-brown latosols are the dominant soil types in the area [77]. Climate, topography and geological condition have resulted in a unique vegetation distribution across the IZ.

For Sri Lanka, the IZ encompasses 13.2% or about 1.2 million ha of the total land mass. Of the total land, approximately 221,977 ha are covered by forest. The vegetation of the IZ is mostly made up of semi-mixed evergreen forest. These forests have a low proportion of deciduous species which make them essentially evergreen. However, deciduous species are mostly found in the forest canopy of the southeastern and northwestern areas of the IZ, so those forests are more deciduous or semi-evergreen in character than those of the central and northern parts [78]. The most dominant plant families in these forests are Anacardiaceae, Euphorbiaceae, Moraceae, and Sapindaceae [79]. These forests provide numerous NTFPs such as edible

products (e.g., fruits, nuts, leafy vegetables, yams, and flowers), rattan, bamboo, medicinal products, bee honey, agricultural by-products (e.g., stalks, green manure, and roping materials), and fodder. A study by Liyanaarachchi [80] revealed that 65–75% of the households in the IZ are reliant on the forest for their daily needs.

The IZ runs across nine out of the twenty-five administrative districts in the country—including larger portions of the Kurunegala, Badulla, and Monaragala districts as well as portions of the Matale, Puttalam, Hambantota, Kandy, Ratnapura, and Nuwara Eliya districts. Based on household income from an expenditure survey conducted by the Department of Census and Statistics, the highest numbers of people who were below the poverty line were recorded in the Kandy and Ratnapura districts of the IZ [81]. Moreover, a study by Menike [82] highlighted three districts in the IZ, namely Badulla, Monaragala, and Ratnapura, as the most poverty-stricken environments. Similar to many of the other parts of the country, the majority of the IZ population live a rural, agricultural subsistent lifestyle [44]. Paddy farming, vegetable cultivation, and shifting (i.e., Chena crop) cultivation are the main farming activities in the IZ. As a result, agriculture is the dominant cause of encroachment of forest lands in the IZ—creating a duality between agriculture output and forest resource use [83]. For example, research has reported that in every season, due to drought, paddy farmers in the Kurunegala district of the IZ have lost 44% of their agriculture income which leads to rural poverty and deforestation [84]. In 2012, in line with the Haritha Lanka Strategy and Action Plan (i.e., green plan) and Caring for Environment National Environmental Action Plan, the FD commenced the SLCFP in the IZ—covering almost all nine districts. For our study, we selected four administrative districts in the IZ; Badulla, Kandy, Kurunegala, and Monaragala, and purposefully selected nine CF sites that were surveyed in these four districts. The forest cover and total population in each CF site are shown in Table 1.

The nine CF sites were selected for four reasons. First, they represent the major vegetation type (i.e., semi-mixed evergreen forest) and are located in the ecological climatic zone, i.e., the IZ, where the majority of CF sites were established. Similarities in vegetation type create the likelihood for species diversity and regenerative patterns [85], hence, these represent similarities with other CF sites for comparability, association, and measure. Second, a large number of forest-dependent people were recorded in these CF sites. Since forest dependency by its inhabitants is high, accompanied by a lack of quantitative data, these sites are of prime importance (e.g., regarding whether they meet inhabitants' daily needs) [80]. Third, these natural forests are heavily susceptible to forest degradation and deforestation due to forest fires, grazing, illegal felling, and encroachment of agriculture. Fourth, recent broad-based studies that encompass parts of the CF sites indicate an increased impact by invasive exotic species, in particular

**Table 1. Forest cover and total population the nine CF sites.**

| District | Name of the CF site (Grama Niladhari Division) | Name of the forest | Extent of the forest (ha) | Total population in CF site |
|---|---|---|---|---|
| Kandy | Bambarabedda | Bambarabedda Waliketiya Mukalana Forest | 69 | 450 |
| Kandy | Wegala | Galkanda natural forest | 60 | 512 |
| Monaragala | Hawanarawa | Hawanarawa natural forest | 50 | 1,050 |
| Kurunegala | Aludeniyaya | Rakaula natural forest and plantation | 900 | 172 |
| Kurunegala | Seeradunna | Dolukanda natural forest | 7,713 | 1,032 |
| Badulla | Dunukewala | Dunukewala natural forest | 237 | 97 |
| Badulla | Gedaboyaya | Gedaboyaya natural forest | 50 | 352 |
| Badulla | Walasgala | Walasgala aluyatawala natural forest | 70 | 1,889 |
| Badulla | Kinniyarawa | Madigala natural forest | 300 | 1,320 |

Source: DCS (2016) [81].

*Lantana camera* (Hinguru), *Eupatorium odoratum* (Podisinghomaran), *Pennisetum polysta-chion* (Mana), *Panicum maximum* (Guinea grass) and *Imperata cylindrical* (Illuk) [76, 86]. These invasive species are abundant across all types of forests in the IZ.

## Data collection

The study followed a semi-experimental, BACI design method [87]. Several studies indicate that BACI designs are appropriate to use for an effectiveness assessment of different program types [88, 89]. Bowler et al. [30] highlighted the BACI design is particularly suited for the assessment of impacts of community forest management programs. Specifically, in the BACI method, an impacted system is compared to a control system before and after a treatment is initiated, without specifying a mechanistic pathway. Here, the BACI framework can be utilized to investigate whether changes in forest condition are likely associated with the implementation of the CF program. Comparatively, forest areas currently under the CF program were regarded as treated (i.e., after treatment), while those that are still managed by the FD were taken as controlled (i.e., before treatment). Also, annual data from 2012 were considered as before the CF program, while from 2018 they was considered after. Following the studies of Hetherington and Willard [90], Malimbwi et al. [91], Mwase et al. [92], Gobeze et al. [37], Obiri et al. [93], Phiri et al. [94], and Chinangwa et al. [34], sample plots and forest inventory techniques were used to collect data (Fig 3).

The total number of sample plots per stand was determined by the uniformity and size of the stand [95]. The scale of the forest area in some CF sites (i.e., Bambarabedda and Wegala districts) is comparatively small; thus, utilizing the operation guideline for community forest management [76] and Vianna and Fearnside [43], a total of 180 plots were sampled across the

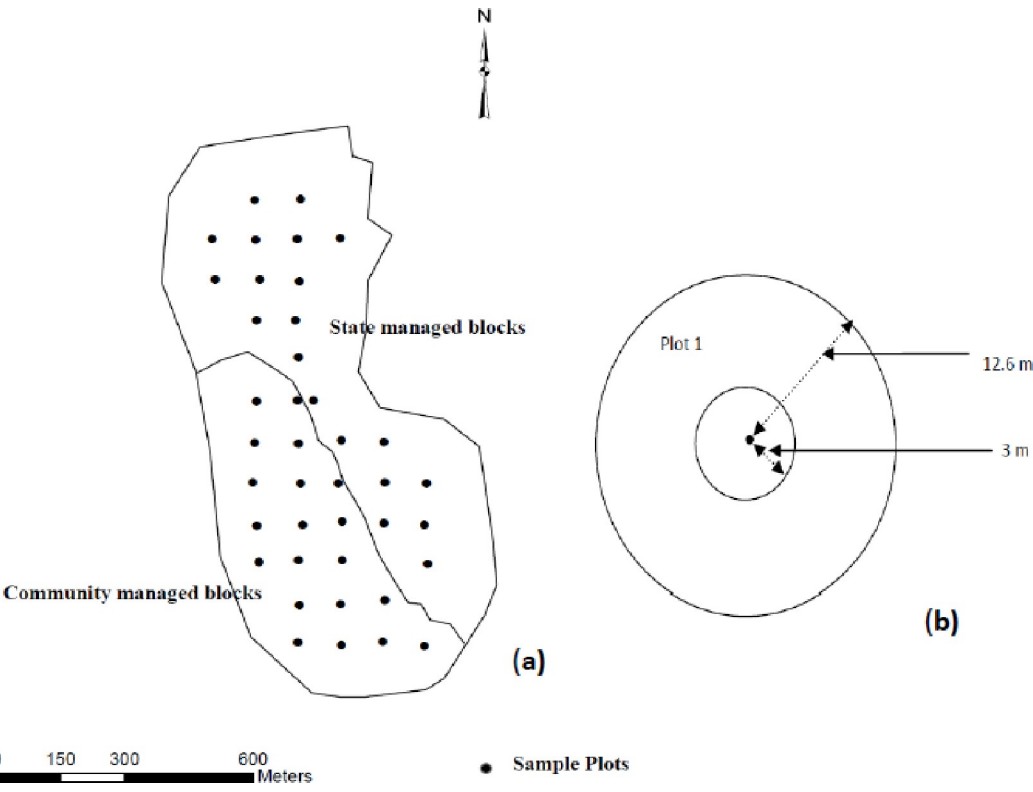

**Fig 3.** Layout of CMB and SMB blocks (a) and plot layout (b).

study site, representing 90 plots in the community- managed blocks (CMBs) and 90 in the state-managed blocks (SMBs). The locations of the sample plots were established on a 100 m by 100 m grid prior to fieldwork. Due to the undulated rocky landscape, these locations were adjusted in the field to avoid obstacles (i.e., rocks). The grid distance was chosen to ensure that plots could be established within the CMBs and SMBs at a minimum distance from the forest boundary (i.e., 100 m). A close-up map of the layout of CMB and SMB blocks in a Bambara-bedda CF site is shown in Fig 3.

Circular plots at 12.6 m radius (i.e., 500 m$^2$) were set as the main plots to inventory standing trees which were greater than a diameter at breast height (DBH) of 5 cm, i.e., lianas, stumps, poles, and felled trees. For saplings (i.e., DBH < 5 cm and height > 1.5 m) and seedlings (i.e., DBH < 5 cm and height < 1.5 m), an inner 3 m radius subplot was set [37]. In addition, invasive species recorded in 3 m radius subplots were counted as a percentage of ground cover [43]. The identification of invasive species used the definition adopted by the Convention of Biological Diversity (CBD). According to the CBD, species "whose introduction and/or spread outside their natural past or present distribution threaten biological diversity are called invasive species" [96]. Invasive species were not counted as saplings or seedlings.

During the study, common names of all plant species were identified with the help of forest field assistants and knowledgeable individuals from the CF sites. Variables in Table 2 were used as indicators to measure the alteration of forest condition with respect to the implementation of the CF program. Indicators were applied to gather and organize information in a manner that made it straightforward to conceptualize, evaluate, and implement the data collection [69, 70]. The plant diversity and composition-based indicators were retained as most valuable [71, 72]. The reference column in Table 2 notes several scientific studies that have applied similar human disturbance-based indicators to measure forest condition [34, 73, 74].

During the study, the 2012 data were collected from inventory data recorded by FD, and the 2018 data were collected by returning to the same sample plots using their recorded GPS locations. In addition, range forest management plans [97–99] and forest offenses record books [100] were used to verify the data on forest management activities (i.e., boundary demarcation and fire-belt establishment) as well as human activities (i.e., forest fires, encroachment, tree felling, and grazing) throughout the nine CF sites. Evidence of natural and human disturbance that occurred before and after the CF program was observed in the main plots of the CMBs and SMBs using parameters from Table 2. The numbers of disease-infected and pest-attacked trees were counted as natural disturbances while the other indicators (i.e., tree

**Table 2. Variables used in assessing impact of the CF program on forest conditions.**

| Parameter | Counting status | Reference |
|---|---|---|
| Number of trees | Main plot | Blomley et al. [73], Gobeze et al. [37] |
| Number of seedlings | Subplot | Gobeze et al. [37], Måren and Sharma [74] |
| Number of saplings | Subplot | Gobeze et al. [37], Måren and Sharma [74] |
| Percent cover of invasive plants | Subplot | |
| Number of tree stumps | Main plot | Blomley et al. [73], Måren and Sharma [74] |
| Number of felled trees | Main plot | |
| Number of lopped trees | Main plot | Chinangwa et al. [34] |
| Number of disease-infected trees | Main plot | |
| Number of pest-attacked trees | Main plot | |
| Land encroachment plots | Main plot | |
| Grazing patches | Main plot | |
| Presence of fire | Main plot | Gobeze et al. [37] |

stumps, felled trees, lopped trees, land encroachment plots, grazing patches, and presence of fire) were counted as human disturbances. According to the Forest Conservation Ordinance, encroachment plots (Table 2) is a count of the number of locations that violate the property rights of the state by building on or extending a structure (clearing land, digs a trench, construction of hut etc.) [101]. Invasive species found throughout Sri Lanka have purposefully been imported and introduced within the horticultural, agricultural, and forestry sectors and are distributed in natural, agricultural, and human settlements. As such, the percentage of invasive plants is included in both natural and human disturbances.

### Data analysis

STATA version 13 software was used to analyze the data. Simple descriptive statistics (i.e., average and percentage) were used to summarize the forest condition parameters.

Woody species diversity was calculated using the Shannon diversity index, Eq (1).

$$H = \sum_{i=1}^{s} - (P_i * lnP_i) \tag{1}$$

Where: "H" = Shannon diversity value, $P_i$ = fraction of the entire population made up of species "i", "s" = number of species encountered, $\Sigma$ = sum from species 1 to species "s", and "ln" = natural log.

A difference in differences (DID) coefficient was used to assess the impact of the CF program by the mean of the indicators for forest condition. The DID model determined the effect of a specific treatment (e.g., large-scale program implementation) by comparing the alteration in outcomes over time between a population which joined the program (i.e., the treatment group) and a population that did not (i.e., the control group) [102]. DID uses longitudinal data of control and treatment groups to gain a suitable counterfactual to analyse a causal effect [103]. The study used Eq (2) to calculate the DID-based model.

$$Y = \beta_0 + \beta_1 D^{post} + \beta_2 D^{Tr} + \beta_3 D^{post}D^{Tr} + \varepsilon \tag{2}$$

Where: "Y" = observed outcomes of the variables in Table 1, β = DID coefficient estimate, $D^{post}$ = time dummy (i.e., 1 = after introducing the CF program), $D^{Tr}$ = treatment group dummy (i.e., 1 = the CF site), $D^{post}D^{Tr}$ = time * treatment interaction, and ε = error term.

During the study, the same variables (variables in Table 1) within two groups (control and treatment) were collected in each period (2012 and 2018). Then by applying the DID method, the average gain in the control group was subtracted from the average gain in the treatment group. This removes the biases in the CF program comparisons between the treatment and control groups. The end result of the DID estimate indicates whether the CF program had an influence (positive or negative) on selected variables (i.e., woody species diversity, tree density, human disturbances).

## Results

### Impacts of the CF program on woody species diversity

A total of 127 woody species (i.e., trees, shrubs, and lianas) occupying 28 families were identified in the nine different forest sites in the IZ (S1 Table). The woody species consisted of 102 species of trees, 19 species of woody shrubs, and six species of woody lianas. Out of the 127 woody species, none are classified as critically endangered; however, *Diplodiscus verrucosus*, *Diospyros chaetocarpa*, and *Miliusa tomentosan* species are classified as endangered and *Antidesma thwaitesianum*, *Canarium zeylanicum*, *Woodfordia fruticosa*, and *Cinnamomum zeylanicum* are classified as vulnerable, according to current conservation records [104, 105]. The

**Table 3. Woody species diversity in semi-mixed evergreen forest.**

| Description | Before the CF program | | After the CF program | |
|---|---|---|---|---|
| | SMBs | CMBs | SMBs | CMBs |
| Number of woody species | 126 | 63 | 127 | 67 |
| Shannon diversity value | 4.47 | 3.65 | 4.47 | 3.76 |

five most abundant woody species recorded in the study sites were *Glycosmis pentaphylla*, *Mallotus philippensis*, *Bauhinia tomentosa*, *Grewia damine*, and *Phyllanthus polyphyllus*. In terms of economically valuable woody trees, *Pterospertmim suberifolium* (timber), *Phyllanthus emblica* (fruits), *Drypetes sepiaria* (fruit), *Manilkara hexandra* (fruits), *Terminalia bellirica* (medicine/fruits), *Cassia auriculata* (flowers) were found as dominant species. From the identified woody species, almost all the species were recorded in the SMBs, while only 67 were recorded on the CMBs (Table 3). The SMBs had a higher and constant Shannon diversity value for the woody trees (i.e., 4.47) than the CMBs (i.e., 3.65 and 3.76), respectively. However, as our study shows woody species diversity increased after the CF program was implemented. The DID coefficient estimate for the Shannon diversity value had a positive value of 0.11. This denotes that the CF program, by itself, positively correlated with species diversity even though it was not significant.

In addition to the statistical analysis, forest management data archived in respective Range Forest Offices (i.e., Hunnasgiriya, Teldeniya, Mahiyangana, Siyabalanduwa, and Kurunegal) where CF sites were established indicated that the types of species which would be planted in the CMBs were preferentially chosen by CF members [100]. Community members preferred plant species which offered economic value. Five species were used (i.e., *Phyllanthus emblica* for fruit and *Tectona grandis*, *Khaya senegalensis*, *Michelia champaca*, and *Pterospertmim suberifolium* for timber), all of which have a high demand in the local market.

## Impacts of the CF program on trees, saplings, and seedlings density

Results show that before the CF program the number of trees per plot varied, ranging from 2 to 64 with a mean of 24.8 in the SMBs and from 1 to 27 with a mean of 12.4 in the CMBs. After the CF program, the number of trees per plot ranged from 2 to 66 with a mean of 25.7 in the SMBs and from 2 to 30 with a mean of 14.1 in the CMBs (Table 4). Our findings indicate that tree density per plot trended upward over time. However, tree density in the CMBs was significantly less (i.e., p < 0.001). The DID coefficient estimate showed the CF program itself increased tree density per plot but did not indicate any significant difference.

**Table 4. The DID estimate of trees, saplings, and seedlings density per plot in the CMBs and SMBs.**

| Variable | CMBs[†] | | SMBs[†] | | DID estimation result | |
|---|---|---|---|---|---|---|
| | Before | After | Before | After | Coefficient | p > (t) |
| Trees | 12.4 | 14.1 | 24.8 | 25.7 | 0.788 | 0.788 |
| Saplings | 2.6 | 5.5 | 2.8 | 3.4 | 2.322*** | 0.000 |
| Seedlings | 18.6 | 88.3 | 14.6 | 26.8 | 57.611*** | 0.000 |

† Mean total; level of significance

* p < 0.05

** p < 0.01

*** p < 0.001 (Overall: R-squared = 0.3220, Prob > F = 0.0000).

Indicators of forest regeneration (i.e., saplings and seedlings) showed different trends between the two forest management systems. The sapling density was comparatively lower than the tree and seedling density in both the CMBs and SMBs. For example, before the CF program, in the CMBs tree density was about five times and seedling density about seven times higher than sapling density, while in the SMBs tree density was about nine times and seedling density five times higher than sapling density. The most commonly occurring sapling species in the SMBs was *Glycosmis pentaphylla* while *Mallotus philippensis* and *Phyllanthus emblica* were the most commonly occurring in the CMBs. The DID coefficient estimate showed a significant positive correlation (i.e., p < 0.001) between the sapling density and the CF program.

In regards to the seedling population, a total of 72 seedling species apart from the invasive species were recorded in the nine CF sites. The five most commonly occurring seedling species in the CMB were *Cassia tora*, *Agerotum conyzoides*, *Vernonia cinereal*, *Stachytarpheta indica*, and *Ocimum tenuiflorum*. Out of these five species, only one species (i.e., *Agerotum conyzoides*) was recorded in the SMBs. Seedlings such as *Clausena indica* and *Streblus taxoides* were commonly recorded throughout the SMBs. Our results indicated that the seedling density per plot in the CMBs gradually increased over time, while the DID coefficient estimate showed a significantly positive relationship (i.e., p < 0.001) between the seedling density and the CF program.

## Impacts of the CF program on the presence of natural and human disturbances

Our results found that before the CF program, human disturbances such as tree stumps, lopped trees, land encroachment plots, grazing patches, and occurrence of forest fires were higher in the CMBs than in the SMBs (Table 5). During the field survey we observed more human disturbances (i.e., stumps, lopped and felled trees, and encroachment patches) near the boundary of the forest reserve and closer to settlements, after which, this gradually declined moving away from the settlements.

The relationship between the CF program and some prominent human disturbance indicators (i.e., tree stumps and felled trees) showed a negative correlation, indicating human disturbances were reduced after the implementation of the CF program, but the relationship was

**Table 5. The DID estimate of natural and human disturbances per plot in the CMBs and SMBs.**

| Variable | CMBs[†] | | SMBs[†] | | DID estimation result[‡] | |
|---|---|---|---|---|---|---|
| | Before | After | Before | After | Coefficient | P > (t) |
| Tree stumps | 2.95 | 2.94 | 1.58 | 1.96 | -0.410 | 0.337 |
| Felled trees | 0.57 | 0.55 | 0.46 | 0.63 | -0.201 | 0.256 |
| Lopped trees | 0.48 | 0.23 | 0.21 | 0.62 | -0.674 | 0.000*** |
| Land encroachment plots | 0.34 | 0 | 0.15 | 0.08 | -0.248 | 0.000*** |
| Grazing patches | 0.51 | 0.03 | 0.15 | 0.35 | -0.683 | 0.000*** |
| Presence of fire | 0.47 | 0.01 | 0.23 | 0.23 | -0.272 | 0.000*** |
| Invasive plants (%) | 44.6 | 7.4 | 14.6 | 25.6 | -48.22 | 0.000*** |
| Disease infected trees | 0 | 0 | 0 | 0 | 0 | |
| Pest-attacked trees | 0 | 0 | 0 | 0 | 0 | |

† mean total

‡ level of significance

* p < 0.05, ** p < 0.01

*** p < 0.001 (Overall: R-squared = 0.1223, Prob > F = 0.0000).

**Table 6. Invasive species recorded in the CMBs and SMBs according to the IUCN (2019).**

| Species name | Abundance | Morphology | Occurrence |
|---|---|---|---|
| *Lantana camera* | high | broad leaved bush | CMBs and SMBs |
| *Eupatorium inulifolium* | low | broad leaved bush to tree | SMBs |
| *Eupatorium odoratum* | high | broad leaved bush to tree | CMBs and SMBs |
| *Tithonia diversifolia* | medium | broad leaved bush | CMBs and SMBs |
| *Ageratum riparium* | low | broad leaved herb to bush | SMBs |
| *Ageratum conyzoides* | medium | broad leaved herb | CMBs and SMBs |
| *Clusia rosea* | low | broad leaved liana to bush | SMBs |
| *Clidemia hirta* | low | broad leaved herb | SMBs |
| *Micania micrantha* | medium | broad leaved liana | CMBs and SMBs |
| *Mimosa invisa* | low | broad leaved liana | CMBs and SMBs |
| *Pennisetum polystachion* | high | grass | CMBs and SMBs |
| *Panicum maximum* | high | grass | CMBs and SMBs |
| *Imperata cylindrica* | high | grass | CMBs and SMBs |

† Red list of threatened species using the Global Invasive Species Database [106].

statistically insignificant. In contrast, it was found that after the CF program, the number of tree stumps, felled trees, and lopped trees had increased in the SMBs. For the land encroachment plots, grazing patches and presence of fire, the CF program had a negative correlation with a statistically significant relationship (i.e., p < 0.001). During the study we did not observe any pest-attacked or disease-infected trees in the selected forests as well as no record of either in the Range Forest Offices (Table 5).

A total of 13 invasive species were recorded in selected semi-mixed evergreen forests (Table 6). Of these, *Lantana camara*, *Mimosa pigra*, and *Imperata cylindrica* are on the "100 of the world's worst" invasive species list of the Global Invasive Species Database [104, 106]. The invasive species consisted of grasses, broad-leaved bushes, broad-leaved herbs, broad leaved lianas, and broad-leaved trees. Out of 13 species, *Eupatorium inulifolium*, *Clusia rosea*, and *Clidemia hirta* were recorded only in the SMBs. *Lantana camera*, *Eupatorium odoratum*, *Pennisetum polystachion*, *Panicum maximum*, and *Imperata cylindrica* were the five most widespread invasive species. These five species are also recorded as the most significant invasive species Sri Lanka-wide [107].

According to our observation, before the CF program, invasive plants occupied 44.6% and 14.6% of ground cover in the CMBs and SMBs, respectively (Table 5). The DID estimates for the percentage of invasive species indicated that the CF program had a significant negative correlation (i.e., p < 0.001) with regards to the percentage of invasive species study-wide. We found that the percentage of invasive species in the SMBs had nearly doubled in 2018 from 2012. On the other hand, in the CMBs, the amount was reduced almost six-fold during the same period.

## Discussion

In regards to using the Shannon diversity index, we found higher values for woody species in the SMBs versus the CMBs. High values of diversity represent more diverse communities. Ifo et al. [108] stated that less species diversity indicates an area is dominated by a single or few species. Our study revealed that the CF program increases woody species diversity, although it was not significant. Lambrick et al. [109], in their analysis of the effectiveness of CF in Prey Long Forest, Cambodia, revealed that results of CF management did not affect woody species

diversity to a significant degree if the sites were implemented two and five years prior to sampling. This is similar to our study which had a five-year sampling period. According to the operation guidelines for CF management in Sri Lanka, all forest blocks handed over to the community were marginal lands with very few scattered valuable trees [45, 76]. Similarly, other studies noted that CF management had been implemented in degraded forests [30]. Furthermore, our results also found that, the CMBs were located closer to villages than the SMBs. In some studies, a shorter distance from the forest to the settlement, identified as a variable, strongly influences the composition of the forest [110].

Due to these differences, tree density and other forest condition measuring variables in the two types of blocks were not equal at the start of the CF program (Table 4). Even though the CF program increased tree density, we could not observe a significant degree of difference since this is dependent on pre-existing forest conditions. The results were similar to the findings reported by Chinangwa et al. [34] in which pre-existing forest conditions affected the impact of the CF program. Moreover, our findings are also in line with Lambrick et al.'s [109] assessment that found valuable tree species were more abundant in control sites (i.e., SMBs) than in the CF sites—most likely since they are located closer to the core of the forest.

In addition, we learned that community members selected trees, playing a major role in maintaining woody species diversity and density in the CMBs. From several studies, it has been evident that people interested in managing forests did so only when they could get both monetary and non-monetary benefits [35, 111, 112]. We found a similar result. While selecting the species in the CMBs, community members gave priority to economic-oriented species rather than diversification of the plant community. Chinangwa et al. [34] reported that the composition and diversity of species in CMBs depend on the decision of community members. Similarly, Pandey et al. [113] reported that CF contributed to high tree species diversity where forest management communities have interests in multiple species, such that most community user groups throughout Nepal leaned toward timber trees that yielded high economic value. As a result, it was observed that CF sites had less tree density than SMBs and national park forests.

Since agriculture is the main source of income in the studied area, it had a very significant effect on the sapling density. For instance, saplings were used as supporting stalks for vegetable (i.e., bean and tomato) cultivation. Hence, before the intervention of the CF program, a smaller number of saplings were recorded in the CMBs that were located near settlements. In addition to this, less presence of saplings is often correlated with higher numbers of full-grown trees. This is also evident in the higher number of tree counts observed in the SMBs. Our study revealed that the CF program by itself increased sapling density significantly. We also found higher sapling density in the CMBs than in the adjacent SMBs. Higher sapling density within the CMBs suggests a more successful enrichment planting program throughout the CF sites. Moreover, the saplings of CMBs consisted of high-value fruit crops or timber tree species, while SMBs were comprised of saplings of dominant woody trees in the specific plots. In some CMBs, more than 70% of saplings were associated with valuable fruit crops. As an example, the data from the Range Forest Office [97] at Hunnasgiriya reported that 5,000 seedlings of *Phyllanthus emblica* were planted on the CMBs during the CF program. At the time of data collection most of the seedlings had grown to the sapling stage. In plots where the largest numbers of saplings were recorded, the saplings were largely *Phyllanthus emblica*. It is argued that *Phyllanthus emblica* fruits play an important role in rural household incomes, especially during the dry period, for purchasing household goods, so they are widely planted by the FD [16]. However, the proportion of the saplings of fruit crops and timber trees was determined by the preference of the community members. Seedlings bloom when the tree canopy is open or tree density is minimal due to less competition for light. Removal of invasive plants encourages

seedling growth as it reduces competition for nutrients, water, and light. We found higher seedling density in the CMBs versus the SMBs. Our findings were consistent with several similar studies in other countries. For example, a study from the Preong Forest in Cambodia [109] revealed there were more regenerating stems per meter on the CF sites than the state-managed ones. Similarly, studies in Tanzania [114] and Ethiopia [37, 115] revealed that after PFM was implemented, significant improvement in seedling and sapling density was present on the PFM sites versus the non-PFM sites. Another positive development was identified by Church [116], who found that tree growth, regeneration, and ground coverage had increased and improved due to the CF program. A study by Miah et al. [117] revealed that after the intervention of PFM, forest cover and forest natural regeneration increased in wet semi-evergreen forests in the Bandarban Hill District of Bangladesh. Unlike these studies, our findings were contrary to the findings of Chinangwa et al. [34]; in the case of the Zomba Malosa Forest Reserve in Malawi, they found that the mean density per plot for seedlings and saplings were significantly higher in the state-managed forest than in the co-managed forest blocks. Obiri et al. [93] also reported that high density seedlings and saplings were observed in SMBs due to appropriate silvicultural practices and adequate enforcement of laws and regulations.

Community dependency on forests was reflected in the forest plot sampling under both management systems. Prior to the intervention of the CF program, the CMBs showed more signs of disturbance, as noted by a higher number of tree stumps, felled trees, lopped trees, and higher numbers of grazing patches. However, after the intervention of the CF program those disturbances were drastically decreased in the CMBs and much has been recorded in the SMBs. This could suggest that stricter management and regulatory monitoring in the CMBs resulted in illegal exploitation in the SMBs, especially lopping tree branches for plucking wild fruits (e.g., *Phyllanthus emblica*) and felling trees for fuel wood and poles (i.e., used as supporting material in construction). Consistent with our findings, UN-REDD [83] specified that 64% of forest loss in Sri Lanka occurred in state-managed, dense forest areas while 36% occurred in open forest areas. Our findings recorded fewer human disturbances in the CMBs than the SMBs, confirming the UN-REDD [83] report. Several similar studies in other countries confirm this as well, for example in India by Somanathan et al. [118], Mexico by Bray et al. [4], and Tanzania by Persha and Blomley [23]. On the other hand, our results indicated that nine natural forests in CF sites exhibited a healthy plant population. Note, lichens are significant features of tropical rainforest and evergreen forests and we did observe lichen growth on branches and trunks of most trees in both the CMBs and SMBs. As they are not harmful we did not include them as a disease.

According to the data from the Range Forest Offices [97–99], forest fires were identified as a key threat during dry spells. Several human activities such as shifting cultivation, grazing animals, and carelessness were other major causes of forest fires. Most of the forest fires were started by nearby villages and spread into core parts of the forest [100]. It was found that as a result of the CF program forest boundaries were clearly marked and a fire-belt constructed in the CMBs. According to the records of the United Nations Framework Convention on Climate Change, nearly 2% of newly planted forest areas in Sri Lanka are burnt annually. On average, the total area burnt by forest fires ranges from 119 ha to 323 ha per year [119]. Statistics from the Department of Forest Conservation states that nearly 900 ha of forest land was burnt in 2012 with that number steadily increasing on an annual basis [95]. The Food and Agriculture Organization of the United Nations highlighted that community-based fire management programs are crucial to preventing and controlling destructive forest fires [120]. Our study revealed that the CF program significantly reduced the occurrence of forest fires in both management blocks. As mentioned in the results, the CMBs were located closer to human settlement whereas the SMBs were located in the core of the forest reserves. Therefore, constructing

a fire-belt in the CMBs indirectly helped reduce fire occurrence in the SMBs. However, a fire belt maintained by CF is ineffective against fires that start inside the state forest or spread from other adjacent settlements away from CF sites. Similar scenarios were also found in the CF program implemented in Nilgala, Sri Lanka [46]. To protect the Nilgala Forest Reserve from fire, community members planted fire-resistant species and established fire-belts surrounding the CF sites. In addition, they hired vigilant committees to patrol both the CF and SMBs throughout the Nilgala Forest Reserve [44, 46]. Our results also support earlier findings by Pandey et al. [113], who found that CF resulted in fewer forest fires and more activities in fire prevention and control.

Records of the Range Forest Offices reported that in agreement with the FD and community members, the FD authorized the planting of cash crops in the CMBs [97–99]. Due to enrichment planting and cash crop cultivation, most of the invasive plants were uprooted by community members. However, *Tithonia diversifolia*, used as a green manure in paddy fields, was kept in the forest without eradication. Continued monitoring and uprooting by community members resulted in a percentage decline of invasive plants in the CMBs. Studies by Flory and Clay [121, 122] reported that hand-weeding or uprooting was most effective in removing plant invasion. Moreover, Borokini and Oluwafemi [123] suggested that community participation should be strongly incorporated into intensive and regular monitoring and management of invasive plant species in Nigeria. Unfortunately the FD is limited in both human and financial resources and thus is unable to monitor and eradicate invasive species effectively [124]. Consequently, the percentage of invasive species in the SMBs increased over time. Our findings were also in line with those of Khadka [125] who conducted an assessment of perceived effects and management challenges of *Mikania micrantha* invasion in Chitwan National Park in Nepal. He concluded that the spread of invasive species was mainly attributed to the lack of forestry management. Nonetheless, in accordance with Meijer et al. [126], our results confirmed that CF facilitates forest management and contributed to maintaining the natural quality of the forest reserve.

In most developing countries including Sri Lanka, a shortage of resources and poor infrastructure have often resulted in a lack of effective state forest management. Our results indicated that transferring management rights and responsibilities to local people encourages them to actively manage the forest, resulting in both ecological and economic benefits. Understanding the changes in forest condition of community-managed forests in general, and natural forests in particular, is quite important not only for conservation and sustainable use of the forest but also for rural development. For example, the positive impact of invasive species control in CMBs will provide useful guidance for FD to undertake policy reform to establish formal systems of community-based invasive species management, which should refine existing legal frameworks to make them more effective.

## Conclusion

Much of the research on the CF program so far has been focused on the CF impacts on rural livelihood and the environment. This research broadens the discussion by incorporating an analysis of the impacts of the CF program on semi-mixed evergreen forests, distributed in tropics and subtropics throughout the world. As such, the study explores the dominant vegetation in the IZ of Sri Lanka where the majority of the CF sites are established. We analyzed the impacts on the dynamics of forest condition through comparative studies via controlled and experimental design applied in nine different natural forests in the IZ. Through this analysis, it was found that the CF program increases forest regeneration (i.e., sapling and seedling density) and reduces invasive species and human disturbance to a significant degree. Findings directly

suggest that increasing access to community-based invasive species management may positively influence the condition of the forest. This research demonstrates the impact of the CF program on forest condition using two-stage study data (i.e., before and after) within a short period of time. The study does not provide details of impact changes over the long term; thus, we recommend that more empirical studies be conducted to evaluate the changes in forest condition over a longer time series.

This research demonstrates that the impact of CF depends on pre-existing forest conditions. For example, a greater amount of regeneration occurred when tree density was low (i.e., in absence of larger trees prior to the implementation of the program). As a result, if a community decides to plant fewer species the results will most likely mean less species diversity and density [61, 71, 72]. In summary, the research can be framed around the concept that the outcome of CF varies with community member understanding and decision-making. To this effect, designing and implementing management strategies to address the impact of the CF program must put pre-existing conditions of the forest and decisions of the local community into consideration when moving towards policy improvements and best practice strategies.

## Supporting information

**S1 Table. List of woody species recorded in semi mixed evergreen forest in the intermediate zone.**
(DOCX)

## Acknowledgments

The authors gratefully acknowledge U. Dammage, N.T.P. Karunaratne, A. Kirindigoda, A.H.S. Nissanka, P Chinthaka, B Bulathwatta, and all the forest officers from the Department of forest Conservation, Sri Lanka for their cooperation and support during the collection of data.

## Author Contributions

**Conceptualization:** E. M. B. P. Ekanayake, Yi Xie.

**Formal analysis:** E. M. B. P. Ekanayake, Yi Xie.

**Funding acquisition:** Yi Xie.

**Investigation:** E. M. B. P. Ekanayake.

**Methodology:** E. M. B. P. Ekanayake, Yi Xie.

**Resources:** E. M. B. P. Ekanayake.

**Supervision:** Yi Xie.

**Validation:** E. M. B. P. Ekanayake, Yi Xie.

**Writing – original draft:** E. M. B. P. Ekanayake, Yi Xie.

**Writing – review & editing:** G. T. Cirella.

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
