## [Decision Letter · Decision Letter 0]

23 Jun 2020

PONE-D-20-13177

Impacts of community forestry on forest condition: Evidence from Sri Lanka's intermediate zone

PLOS ONE

Dear Dr. Xie,

Thank you for submitting your manuscript to PLOS ONE. After careful consideration, we feel that it has merit but does not fully meet PLOS ONE’s publication criteria as it currently stands. Therefore, we invite you to submit a revised version of the manuscript that addresses the points raised during the review process.

I agree with Reviewer 1, who also reviewed the previously submitted version of this manuscript: this version is a major improvement, and should be suitable for publication after some clarifications and other changes. Both reviewers have provided excellent guidance regarding how you should proceed with your revisions. Please read through their comments carefully and attempt to address all of the points they raised. As suggested by Reviewer 2, you need to focus on two aspects in particular: (1) providing more detail about the FD sites (i.e., the SMBs), including how they relate to the CMBs spatially and how they were selected for this study; and (2) more balance in the discussion with respect to the differing initial conditions of the CMBs versus the SMBs. These are not the only areas that you need to address, but tackling these should enhance the manuscript immensely. In addition, I went through the manuscript and identified some minor editorial and grammatical changes. I have attached an annotated version of the manuscript that documents my suggestions.

We look forward to receiving your revised manuscript.

Kind regards,

Frank H. Koch, PhD

Academic Editor

PLOS ONE

Journal Requirements:

Additional Editor Comments:

Although it wasn't important at this stage in the review process, you will need to ensure that the references and citations conform to the PLOS ONE style guidelines.

Reviewers' comments:

Reviewer's Responses to Questions

**Comments to the Author**

1. Is the manuscript technically sound, and do the data support the conclusions?

Reviewer #1: Partly

Reviewer #2: Partly

2. Has the statistical analysis been performed appropriately and rigorously? 

Reviewer #1: Yes

Reviewer #2: I Don't Know

3. Have the authors made all data underlying the findings in their manuscript fully available?

Reviewer #1: No

Reviewer #2: Yes

4. Is the manuscript presented in an intelligible fashion and written in standard English?

Reviewer #1: Yes

Reviewer #2: Yes

5. Review Comments to the Author

Reviewer #1: SECOND REVIEW

This review of the manuscript “Impacts of community forestry on forest condition: Evidence from Sri Lanka's intermediate zone” (PONE-D-20-13177) constitutes the second revision of the previous version “Community forestry in Sri Lanka: Case study on impacts on forest condition” (PONE-D-19-21562). The authors have revised the study and analysis, attending the main issue concerning its applicability for publication in PLOS ONE journal, mainly the sample size of CF sites for its assessment. In the revised version, the sample increases form being a case study of one CF site of 70 ha, to a total of 9 CF sites with greater spatial coverage of mixed semi-evergreen forest in Sri Lanka. Other issues regarding its redaction and greater detail in the results presented, such as the use and value of seedlings, saplings and trees recorded in sampled plots have also been attended. Thus, there is a major improvement to the manuscript and is now acceptable for publication after some modifications and clarifications in the text that require attention.

In the Introduction it would help to present a more formal and current definition of Community Forest Management (e.g. Charnley and Poe, 2007); what are the criteria in terms of forest resource access, use and rights, land tenure, govt. institutions, etc. and how are these conditions represented in Sri Lanka? In the Methodology section, while the authors have offered a list of specific management interventions undertaken by the FD in the forest before CF, they still do not clearly characterize how management interventions differ (or not) before and after CF implementation within CF and SM forests. Forest conditions are clearly different in CF and SM forests before the treatment, and apparently this has to do with proximity to settlements (e.g. more disturbance and less spp. diversity). Are the exact same management prescriptions applied to CF and SM forest despite contrasting forest conditions and geographies? Is the only difference concerning forest management in CF forest after treatment that of being decided and controlled by the community and not entirely by the FD as before treatment? The impression is that management interventions occur in CF sites (before and after) and not in SM sites (no management or just protection). Please clarify this in the text, or it could be presented in a simple table.

Additionally, in the Methods, Data Collection section of the manuscript, the authors need to state in clearer fashion the data and source used to characterize forest condition and disturbance in plots (CMB and SMB blocks) before (i.e. 2012) and after (i.e. 2018) treatment (i.e. CF). It appears that the 2012 data was provided by the FD and the authors collected the 2018 data returning to the same sample plots both in CMBs and SMBs. Is this the case? Then please state clearly in the manuscript. Moreover, how are the CMBs and SMBs spatially related in a CF site. A close-up map of the layout of CMB and SMB blocks or plots in a representative CF site would help clear this up. The effect of distance is mentioned, although not considered a co-variate, and SMB block are implied as being distant to CMB block, but one does not have an idea of the spatial distribution of blocks being compared (before and after). Are SMBs located in neighboring state managed forests of CF sites or adjacent to CMBs? Are all plots 100 m apart in CMBs and SMBs, please clarify in text and/or map. By the way, the raw data provided seems to only be for CMBs or the treated block there is not a column that indicates control units (SMB) and treated units (CMB), adding to the confusion on this subject. Please make all pertinent data available, in addition to summary statistics, full DID model results as indicated by PLOS Data Policy.

In Results, some more detail on the proportions of useful/valuable species (trees, saplings and seedlings) for fruit and timber in CMB vs. SMB before and after need to be included. This would help a lot in understanding the environmental and social benefits brought by the CF program and strengthen the Discussion. It could be included in text or a separate column in Tables showing BACI results. Moreover, the fitness and significance of DID linear model needs to be reported (see above on data availability). Finally, there a few scattered typos and errors in the text, so a thorough editorial review is also needed for the next revision.

Charnley, S.; Poe, M.R. Community forestry in theory and practice: Where are we now? Annu. Rev. Anthropol. 2007, 36, 301–336.

Reviewer #2: GENERAL: The authors present a semi-experimental BACI design study to determine the ecological impacts of community forests (CF) as compared to governmental controlled (FD) forests in Sri Lanka. Their findings indicate that CF showed a significant increase in sapling and seedling density as compared to state-controlled forests, as well as a decrease in certain human disturbance activities. Although the authors present a nice study that I believe would be suitable and of interest to PLOS ONE readers, there are two main issues of concern at present: the lack of detail on the FD sites and greater balance of discussion on the findings in relation to the starting point of the respective forest conditions. Please see my comments and suggestions below.

INTRODUCTION

P 4 line 71-73, I would suggest that there are stronger citations that might be considered here including Porter-Bolland et al 2012 For Ecol Mang and Bowler et al 2012 Frontiers in Ecol

METHODS

Some grammatical errors - e.g. p 8 line 171 use of the word ‘implication’, p 11 line 233 what are agriculture ‘implies’

P 12 line 253, I am a bit confused by your rationale that drought and the loss of crops can be attributed to ‘lead[ing] to rural poverty via deforestation’. By definition, subsistence farmers are poor. Drought and the loss of a large portion of crops could certainly drive a shift in livelihood practices, eg deforestation, driving people further into poverty (or the corollary, be a ‘safety net’ preventing such), but it would not be what made them poor - that is, they already were!

P 12 My first major issue with your paper is the lack of detail on the FD sites. I understand that the CF sites were purposefully selected as I imagine the FD sites were too? There is, however, no indication of such in the methods. Furthermore, I wonder if there were any attempts to ‘match’ or minimally reduce major differences in the CF and FD sites, i.e. proximity to markets, distance to paved road, human population size, or other characteristics? I would also appreciate greater detail on the FD site locations, like that which is provided for the CF sites in Table 1. My ability to evaluate the robustness of your analysis lies in part on my understanding of the validity of the controls, which is a bit elusive at present.

P 15 There is quite a bit of variation in size of the CFs (50-7,713 ha). How did the FD sites compare? And, I am perplexed why you would sample similarly in areas of such large difference - would this not be particularly important for the large area that may contain rare species or potentially greater variation?

I assume there is no issue in identifying the areas where the study occurred? I would suggest modifying your location map to include distinguishing between CF and FD sites.

P 17 I would appreciate a greater understanding of some of your variables, in particular, the number of disease infected trees, pest-attacked trees, land encroachment plots, grazing patches. For example, what constituted ‘encroachment’? Was there a minimum area that had to be disturbed in a defined manner to qualify as such?

So data was collected in the same 180 plots first in 2013 and then again in 2018? I think this could be made more explicit earlier on in the methods. With the number of trees lopped or felled, how did you ensure that they were not double counted?

RESULTS

P 21 Paragraph beginning, ‘Indicators of forest. . . ‘ I found your emphasis on tree and seed densities being higher than sapling density confusing with your last sentence that emphasizes increasing sapling density with CF implementation. I would also suggest drawing out the difference between CMB and SMB sites more explicitly.

P 23 lines 465-468 This seems quite anecdotal and perhaps more suited to the discussion?

P 24, line 473 Delete ‘however’

DISCUSSION

P 25 line 501 It is here that we learn that the CF were established on ‘marginal lands with very few scattered valuable trees’ as well as generally being located closer to villages than SMBs. I would suggest these are rather critical details in the interpretation of your findings as well as things that should likely be stated earlier on in the manuscript. I also wonder how valid it is to compare the impact of CF using SMBs as a comparison if their ecological starting points are so vastly different?

Nonetheless, I find aspects of your results quite compelling and insightful. In particular, the reduction in invasives and fire occurrence are no doubt due in large part to community effort. åSimilarly, that seedling and sapling densities have increased and certain human activities, decreased. I would suggest greater attention to your selection of the FDs and being more upfront about the differences in CMBs versus SMBs will strengthen the paper and facilitate its publication.

6. PLOS authors have the option to publish the peer review history of their article (what does this mean?). If published, this will include your full peer review and any attached files.

Reviewer #1: No

Reviewer #2: Yes: Nicole Gross-Camp

---

## [Author Response · Author response to Decision Letter 0]

16 Jul 2020

Reviewer 1

Question 1: Is the manuscript technically sound, and do the data support the conclusions?

Comment of Reviewer #1: Partly

Response: We are grateful to the first reviewer’s comment. We have modified the original version with technically sound scientific research data that supports the conclusions. Moreover, our experiment was conducted rigorously, with appropriate controls, replication, and sample sizes.

Question 2. Has the statistical analysis been performed appropriately and rigorously?

Comment of Reviewer #1: Yes

Response: We are grateful to the first reviewer’s comment.

Question 3. Have the authors made all data underlying the findings in their manuscript fully available?

Comment of Reviewer #1: NO

Response: Thank you for the helpful comment. In the current version of the manuscript, the data underlying the findings described in the manuscript are fully available without restriction.

Question 4. Is the manuscript presented in an intelligible fashion and written in Standard English?

Comment of Reviewer #1: Yes

Response: We are grateful to the first reviewer’s comment.

Question 5. 

Comment of Reviewer #1:

This review of the manuscript “Impacts of community forestry on forest condition: Evidence from Sri Lanka's intermediate zone” (PONE-D-20-13177) constitutes the second revision of the previous version “Community forestry in Sri Lanka: Case study on impacts on forest condition” (PONE-D-19-21562). The authors have revised the study and analysis, attending the main issue concerning its applicability for publication in PLOS ONE journal, mainly the sample size of CF sites for its assessment. In the revised version, the sample increases form being a case study of one CF site of 70 ha, to a total of 9 CF sites with greater spatial coverage of mixed semi-evergreen forest in Sri Lanka. Other issues regarding its redaction and greater detail in the results presented, such as the use and value of seedlings, saplings and trees recorded in sampled plots have also been attended. Thus, there is a major improvement to the manuscript and is now acceptable for publication after some modifications and clarifications in the text that require attention.

Response: We are grateful to the first reviewer’s beneficial comments. We have modified the original version and have re-submitted a new version by following all of your comments.

(a) Present a more formal and current definition of Community Forest Management (e.g. Charnley and Poe, 2007).

Response: We are grateful to this comment. The current version of the manuscript is revised by including the more formal and current definition of CF (line 46-49)

(b)What are the criteria in terms of forest resource access, use and rights, land tenure, govt. institutions, etc. and how are these conditions represented in Sri Lanka?

Response: We are grateful to this comment. The manner in which power and rights (i.e., resource access) change from state-managed to community-managed is shown in the conceptual framework (Fig. 1) and line 176-186.

(c) In the Methodology section, while the authors have offered a list of specific management interventions undertaken by the FD in the forest before CF, they still do not clearly characterize how management interventions differ (or not) before and after CF implementation within CF and SM forests. The impression is that management interventions occur in CF sites (before and after) and not in SM sites (no management or just protection). Please clarify this in the text, or it could be presented in a simple table.

Response: We are grateful to this comment. The current version of the manuscript is revised by adding the main differences of management in community and state forests before and after implementing the CF program (line 178-182).

(d) It appears that the 2012 data was provided by the FD and the authors collected the 2018 data returning to the same sample plots both in CMBs and SMBs. Is this the case? Then please state clearly in the manuscript.

Response: We are grateful to this comment. The current version of the manuscript is revised by indicating the data collection method in both 2012 and 2018 (line 338-340).

(e)How the CMBs and SMBs are spatially related in a CF site. A close-up map of the layout of CMB and SMB blocks or plots in a representative CF site would help clear this up. 

Response: We are grateful to this comment. The current version of the manuscript is revised by drawing a layout map which shows the CMBs and SMBs spatially related in the CF site (Fig. 3) 

(f) The raw data provided seems to only be for CMBs or the treated block there is not a column that indicates control units (SMB) and treated units (CMB), adding to the confusion on this subject. Please make all pertinent data available, in addition to summary statistics, full DID model results as indicated by PLOS Data Policy.

Response: The current version of the manuscript is revised by making fully available data indicating control (SMB) and treated unit (CMB).

(g)The fitness and significance of DID linear model needs to be reported (see above on data availability). 

Response: We are grateful to this comment. The current version of the manuscript is revised by adding related references that show fitness and significance of DID linear model to the current study (line 366-368).

(h) Finally, there a few scattered typos and errors in the text, so a thorough editorial review is also needed for the next revision.

Response: We are grateful for the comment. In the current version of the manuscript, the language has been corrected with the help of a professional English editor.

Reviewer 2

Question 1: Is the manuscript technically sound, and do the data support the conclusions?

Comment of Reviewer #1: Partly

Response: We are grateful to the second reviewer’s comment. We have modified the original version with technically sound scientific research data that supports the conclusions. Also, our experiment was conducted rigorously, with appropriate controls, replication, and sample sizes.

Question 2. Has the statistical analysis been performed appropriately and rigorously?

Comment of Reviewer #2: I don’t Know

Response: Thank you for comment. The current version of the manuscript is revised by adding related references that shows fitness and significance of DID linear model to the current study (line 372-374).

Question 3. Have the authors made all data underlying the findings in their manuscript fully available?

Comment of Reviewer #2: Yes

Response: Thank you very much for your comment. In the current version of the manuscript, the data underlying the findings described in the manuscript are fully available and indicating control (SMB) and treated unit (CMB).

Question 4. Is the manuscript presented in an intelligible fashion and written in Standard English?

Comment of Reviewer #1: yes

Response: Thank you for the helpful comment.

Question 5. 

Comment of Reviewer #2: The authors present a semi-experimental BACI design study to determine the ecological impacts of community forests (CF) as compared to governmental controlled (FD) forests in Sri Lanka. Their findings indicate that CF showed a significant increase in sapling and seedling density as compared to state-controlled forests, as well as a decrease in certain human disturbance activities. Although the authors present a nice study that I believe would be suitable and of interest to PLOS ONE readers, there are two main issues of concern at present: the lack of detail on the FD sites and greater balance of discussion on the findings in relation to the starting point of the respective forest conditions. Please see my comments and suggestions below.

Response: We are grateful to the second reviewer’s beneficial comments. We have modified the original version and have re-submitted a new version by following all of your comments.

INTRODUCTION

Comment: P 4 line 71-73, I would suggest that there are stronger citations that might be considered here including Porter-Bolland et al 2012 For Ecol Mang and Bowler et al 2012 Frontiers in Ecol

Response: Thank you for your valuable suggestion. The manuscript has been modified by adding suggested references (line 73).

METHODS

Comment: Some grammatical errors - e.g. p 8 line 171 use of the word ‘implication’, p 11 line 233 what are agriculture ‘implies’ 

Response: We are grateful for this comment. The manuscript has been modified by correcting the word “implementation” (line173). Agriculture implies are the materials extract from the forest and which use as supportive materials in pandals making in some crops (sticks, ropes).

Comment: P 12 line 253, I am a bit confused by your rationale that drought and the loss of crops can be attributed to ‘lead[ing] to rural poverty via deforestation’. By definition, subsistence farmers are poor. Drought and the loss of a large portion of crops could certainly drive a shift in livelihood practices, eg deforestation, driving people further into poverty (or the corollary, be a ‘safety net’ preventing such), but it would not be what made them poor - that is, they already were!

 Response: Thank you for this comment. The manuscript has been modified by correcting sentence (lines 256-258). 

Comment: P 12 My first major issue with your paper is the lack of detail on the FD sites. I understand that the CF sites were purposefully selected as I imagine the FD sites were too? There is, however, no indication of such in the methods. Furthermore, I wonder if there were any attempts to ‘match’ or minimally reduce major differences in the CF and FD sites, i.e. proximity to markets, distance to paved road, human population size, or other characteristics. I would also appreciate greater detail on the FD site locations, like that which is provided for the CF sites in Table 1. My ability to evaluate the robustness of your analysis lies in part on my understanding of the validity of the controls, which is a bit elusive at present.

 Response: Thank you for your valuable comment. In this study we consider the CF site which includes community manage blocks as well as state-managed blocks. The purpose of selecting these 9 CF sites are mentioned in lines 269-280. To make a clear understanding of the distribution of SMB and CMB, a close-up map of the layout of CMB and SMB blocks in a representative CF site (Bambarabedda CF) was included into the new manuscript (Fig. 3).

Comment: P 15 There is quite a bit of variation in size of the CFs (50-7,713 ha). How did the FD sites compare? And, I am perplexed why you would sample similarly in areas of such large difference - would this not be particularly important for the large area that may contain rare species or potentially greater variation? I assume there is no issue in identifying the areas where the study occurred? I would suggest modifying your location map to include distinguishing between CF and FD sites.

Response: Thank you for your valuable comment. To make a clear understanding of the distribution of SMB and CMB in CF sites a close-up map of the layout of CMB and SMB blocks in a representative CF site (Bambarabedda CF) was included into the new manuscript (Fig. 3).

Comment: P 17 I would appreciate a greater understanding of some of your variables, in particular, the number of disease infected trees, pest-attacked trees, land encroachment plots, grazing patches. For example, what constituted ‘encroachment’? Was there a minimum area that had to be disturbed in a defined manner to qualify as such? So data was collected in the same 180 plots first in 2013 and then again in 2018? I think this could be made more explicit earlier on in the methods. With the number of trees lopped or felled, how did you ensure that they were not double counted?

Response: We are grateful for this comment. Forest encroachment means to violate the boundary of state land and clearing the land for any other purpose other than forestry (agriculture, construction, mining, digging etc.). As mentioned in lines 267-300, 303-304 in the revised manuscript, the study followed the inventory techniques. According to the literature I followed in the study (especially FD,2014) there are systems of inventorying the trees without double counting. Therefore during the study, no trees were double-counted.

RESULTS

Comment: P 21 Paragraph beginning, ‘Indicators of forest. . . ‘ I found your emphasis on tree and seed densities being higher than sapling density confusing with your last sentence that emphasizes increasing sapling density with CF implementation. I would also suggest drawing out the difference between CMB and SMB sites more explicitly.

Response: Thank you for your comment. In general, compare to seedling and tree density sapling density is lower in both CMB and SMB. The reason for this is mentioned in line 528-532 in the discussion section of new manuscript. However, the CF program improves the sapling density in CMB but it did not exceed the seedling or tree density. 

Comment: P 23 lines 465-468 this seems quite anecdotal and perhaps more suited to the discussion?

Response: We agree with your comments. The manuscript has been modified by moving the section into discussion part (line 578-582).

Comment: P 24, line 473 Delete ‘however’

Response: We are grateful for this comment. Current version of manuscript modified by deleting however (line 472).

DISCUSSION

Comment: P 25 line 501 It is here that we learn that the CF were established on ‘marginal lands with very few scattered valuable trees’ as well as generally being located closer to villages than SMBs. I would suggest these are rather critical details in the interpretation of your findings as well as things that should likely be stated earlier on in the manuscript. I also wonder how valid it is to compare the impact of CF using SMBs as a comparison if their ecological starting points are so vastly different.

Response: Thank you for your comment. As mentioned in line 500-502 in the original manuscript “all forest blocks handed over to the community were marginal lands with very few scattered valuable trees (FD, 2014; Dissanayake, 2013). The layout map (fig 3) shows how CMB and SMB are located in CF sites. In our study, we found that CMBs were closed to the villages and it was the reason for higher human disturbance and less tree density in CMBs. As we found this result in our study we included this section in the discussion part. In addition, we used the DID model to compare the impact of CF. In DID analysis, comparison groups can start at different levels of the outcome (DID focuses on change rather than absolute levels) (Angrist and Pischke, 2008). This is the main strength of the DID which makes it more suitable for the current study.

Angrist J, Pischke JS. Mostly Harmless Econometrics. Princeton University Press, NJ 2008.

Comment: Nonetheless, I find aspects of your results quite compelling and insightful. In particular, the reduction in invasives and fire occurrence are no doubt due in large part to community effort. åSimilarly, that seedling and sapling densities have increased and certain human activities, decreased. I would suggest greater attention to your selection of the FDs and being more upfront about the differences in CMBs versus SMBs will strengthen the paper and facilitate its publication.

Response: We modified the manuscript by drawing the layout of CMB and SMB and adding the main differences of the state forest which are subjected to implement CF program and which are not (Fig. 3, line 178-182).

---

## [Editor Report · Decision Letter 1]

27 Jul 2020

PONE-D-20-13177R1

Impacts of community forestry on forest condition: Evidence from Sri Lanka's intermediate zone

PLOS ONE

Dear Dr. Xie,

Thank you for submitting your manuscript to PLOS ONE. After careful consideration, we feel that it has merit but does not fully meet PLOS ONE’s publication criteria as it currently stands. Therefore, we invite you to submit a revised version of the manuscript that addresses the points raised during the review process.

I appreciate the time you took to respond to the reviewer’s comments. However, the manuscript is not yet suitable for publication, as you still need to address several important issues:

It doesn’t appear that the individual figure files are in the file inventory for this manuscript. Please ensure that these are uploaded to the Editorial Manager.The references are not formatted for PLOS ONE (i.e., numbered and listed in the order in which they appear in the text). The citations in the text must also be converted to PLOS ONE’s numerical format.Figure 2 should include the jurisdictional boundaries of all nine administrative districts named in the manuscript. The current map is a bit visually confusing.One of the main criticisms from both reviewers was a lack of clarity about the CF sites. It is clearer now that you were actually looking at paired sites in each case, i.e., each site had a community-managed block and state-managed block, which you sampled equally (90 plots in each). Figure 3 is very helpful in this regard. However, you didn’t really address the large difference in forest extent between the sites (as shown in Table 1), as noted by Reviewer #2. Because you sampled using a 100 m X 100 m grid, I assume you only sampled a limited proportion of the larger sites (e.g., Aludeniyaya and Seeradunna), which is fine, but it would be good if you clarified this and defended why you think you didn’t overlook rare species or underestimate variation by doing so. For readers’ benefit, you should also show the locations of the sites on the map in Figure 2. You can represent them as points.You have not yet addressed this comment from Reviewer #1: “In the Methodology section, while the authors have offered a list of specific management interventions undertaken by the FD in the forest before CF, they still do not clearly characterize how management interventions differ (or not) before and after CF implementation within CF and SM forests. The impression is that management interventions occur in CF sites (before and after) and not in SM sites (no management or just protection).” As you indicated, state-managed forests are still subject to FD policies and decision-making, while decision-making in community-managed forests is largely ceded to their associated communities. What does that mean in a practical sense? Are most state-managed forests still being managed actively by FD? For example, do forestry operations take place? As the reviewer suggests, readers may have the impression that no interventions occur in state-managed forest blocks once neighboring blocks transition to CF, but I’m guessing that isn’t the case. Regardless, you need to clarify this in the Methods section. (Also see my related comment #8, below)Reviewer #1 asked you to report the overall fitness and significance of the DID linear model, in addition to the significance of the individual coefficients shown in Table 4. What was the overall p-value and R-squared?You have not fully addressed an important comment from Reviewer #2: “I also wonder how valid it is to compare the impact of CF using SMBs as a comparison if their ecological starting points are so vastly different.” You note one of the most critical differences between the CMBs and SMBs (i.e., by design, forests handed over to communities were marginal lands), but you don’t bring this up until the Discussion section. As suggested by the reviewer, you should present this contrast earlier in the manuscript. The facts that the community-managed forest blocks were all on marginal lands and were closer, on average, to the communities than the state-managed blocks should be acknowledged in the Methods section as well as the Discussion; readers should have this information before they see the results.“Land encroachment plots” appears to be a count variable. You need to better describe the standard for recording an incidence of encroachment. For instance, if someone digs a trench and it accidentally crosses into a forest block by a meter and takes out a sapling or two, is this encroachment?You have not fully provided the data underlying the findings. Table S1 is helpful in that regard but full data availability would include – for each CMB and SMB and for both 2012 and 2018 – values for each of the variables listed in Table 2. Please review the PLOS ONE policies on data availability: https://journals.plos.org/plosone/s/data-availability. Most importantly, PLOS journals require authors to make all data necessary to replicate their study’s findings publicly available without restriction at the time of publication. When specific legal or ethical restrictions prohibit public sharing of a data set, authors must indicate how others may obtain access to the data.

We look forward to receiving your revised manuscript.

Kind regards,

Frank H. Koch, PhD

Academic Editor

PLOS ONE

Additional Editor Comments (if provided):

I have suggested a number of minor editorial and grammatical changes. Please see the attached version of the manuscript file with tracked changes.

---

## [Author Response · Author response to Decision Letter 1]

9 Aug 2020

Academic Editor Comments:

Comment 1

It doesn’t appear that the individual figure files are in the file inventory for this manuscript. Please ensure that these are uploaded to the Editorial Manager.

Response 1

Thank you for the helpful comment. With the current version of the manuscript, three individual figure files are uploaded to the Editorial Manager (Figure 1, 2, and 3).

Comment 2

The references are not formatted for PLOS ONE (i.e., numbered and listed in the order in which they appear in the text). The citations in the text must also be converted to PLOS ONE’s numerical format.

Response 2

We are grateful to this comment. The current version of the manuscript, the references, and citations in the text are converted to PLOS ONE’s format.

Comment 3

Figure 2 should include the jurisdictional boundaries of all nine administrative districts named in the manuscript. The current map is a bit visually confusing.

Response 3

We are grateful to the first reviewer’s beneficial comments. The current version of the manuscript modified by including a map consists of the jurisdictional boundaries of all nine administrative districts (Figure 2).

Comment 4

One of the main criticisms from both reviewers was a lack of clarity about the CF sites. It is clearer now that you were actually looking at paired sites in each case, i.e., each site had a community-managed block and state-managed block, which you sampled equally (90 plots in each). Figure 3 is very helpful in this regard. However, you didn’t really address the large difference in forest extent between the sites (as shown in Table 1), as noted by Reviewer #2. Because you sampled using a 100 m X 100 m grid, I assume you only sampled a limited proportion of the larger sites (e.g., Aludeniyaya and Seeradunna), which is fine, but it would be good if you clarified this and defended why you think you didn’t overlook rare species or underestimate variation by doing so. For readers’ benefit, you should also show the locations of the sites on the map in Figure 2. You can represent them as points

Response 4

We are grateful to this comment. The sample plots used in this study originally were laid by the Department of Forest Conservation following the uniformity and size of the stand. The authors laid the same plot using the recoded GPS location. Therefore, we did not lay plots to overlook rare species or underestimate variation. Moreover, we modified the manuscript by including the map indicating the locations of the sites (Figure 2).

Comment 5

You have not yet addressed this comment from Reviewer #1: “In the Methodology section, while the authors have offered a list of specific management interventions undertaken by the FD in the forest before CF, they still do not clearly characterize how management interventions differ (or not) before and after CF implementation within CF and SM forests. The impression is that management interventions occur in CF sites (before and after) and not in SM sites (no management or just protection).” As you indicated, state-managed forests are still subject to FD policies and decision-making, while decision-making in community-managed forests is largely ceded to their associated communities. What does that mean in a practical sense? Are most state-managed forests still being managed actively by FD? For example, do forestry operations take place? As the reviewer suggests, readers may have the impression that no interventions occur in state-managed forest blocks once neighboring blocks transition to CF, but I’m guessing that isn’t the case. Regardless, you need to clarify this in the Methods section. (Also see my related comment #8, below

Response 5

We are grateful to this comment. The manuscript has been modified by including specific management interventions undertaken by the FD in the state-managed blocks before and after CF (Lines 170-174).

Comment 6

Reviewer #1 asked you to report the overall fitness and significance of the DID linear model, in addition to the significance of the individual coefficients shown in Table 4. What was the overall p-value and R-squared?

Response 6

We are grateful to this comment. The current version of the manuscript modified by adding overall p-value and R-squared (Lines 407-408, 440-441).

Comment 7

You have not fully addressed an important comment from Reviewer #2: “I also wonder how valid it is to compare the impact of CF using SMBs as a comparison if their ecological starting points are so vastly different.” You note one of the most critical differences between the CMBs and SMBs (i.e., by design, forests handed over to communities were marginal lands), but you don’t bring this up until the Discussion section. As suggested by the reviewer, you should present this contrast earlier in the manuscript. The facts that the community-managed forest blocks were all on marginal lands and were closer, on average, to the communities than the state-managed blocks should be acknowledged in the Methods section as well as the Discussion; readers should have this information before they see the results.

Response 7

We are especially grateful to this comment. The current version of the manuscript has been modified by adding the fact that “the community-managed forest blocks were all on marginal lands and were closer to the communities than the state-managed blocks” into the methodology section (Lines 202-204).

Comment 8

“Land encroachment plots” appears to be a count variable. You need to better describe the standard for recording an incidence of encroachment. For instance, if someone digs a trench and it accidentally crosses into a forest block by a meter and takes out a sapling or two, is this encroachment

Response 8

We are grateful to this comment. The current version of the manuscript modified by adding the standard for recording an incidence of encroachment (Lines 330-332).

Comment 9

You have not fully provided the data underlying the findings. Table S1 is helpful in that regard but full data availability would include – for each CMB and SMB and for both 2012 and 2018 – values for each of the variables listed in Table 2

Response 9

We are also very grateful to this comment. Raw data of each variable mentioned in Table 2 were provided in the Excel sheet named “Raw Data” and will be available without restriction from the corresponding author (upon reasonable request).

Additional Editor Comment

I have suggested a number of minor editorial and grammatical changes. Please see the attached version of the manuscript file with tracked changes.

Response

We are grateful to the additional Editor comments and feel the review process has much improved our manuscript. The comments have been extremely beneficial and opportune in bettering our work. The current version of the manuscript has been modified by correcting editorial and grammatical errors. Corrected words are highlighted in red in the file named: “Revised Manuscript with Track Changes”.

---

## [Editor Report · Decision Letter 2]

17 Aug 2020

PONE-D-20-13177R2

Impacts of community forestry on forest condition: Evidence from Sri Lanka's intermediate zone

PLOS ONE

Dear Dr. Xie,

Thank you for submitting your manuscript to PLOS ONE. After careful consideration, we feel that it has merit but does not fully meet PLOS ONE’s publication criteria as it currently stands. Therefore, we invite you to submit a revised version of the manuscript that addresses the points raised during the review process.

The manuscript is nearly suitable for publication but you need to clarify your interpretation of the results with respect to fire. For example, in lines 593-595 you state that forest boundaries were clearly marked and a fire-belt constructed in the CMBs as a result of the CF program. However, in newly added text (lines 170-174) you indicate that the FD did those things, presumably once the blocks were selected for the CF program. It’s important to be clear about that. In lines 601-605, you assert that the CF program reduced the occurrence of forest fires in both CMBs and SMBs. This conflicts somewhat with Table 5 – yes, the DID coefficient for fire is negative and significant, but the values of the fire presence variable for the SMBs are the same before and after (0.23). So, did the CF program really make a difference in terms of fire in the SMBs? (On the other hand, it seems pretty conclusive for the CMBs.)

I also have some minor editorial comments for you to address (see "Additional Editor Comments").

We look forward to receiving your revised manuscript.

Kind regards,

Frank H. Koch, PhD

Academic Editor

PLOS ONE

Additional Editor Comments (if provided):

Line 78 – delete comma after “program”

Line 79 – change “subsistent” to “subsistence”

Line 85 – insert comma after “sacred”

Line 102 – delete “very”

Line 150 – spell out non-timber forest products at this first use

Line 163 – “transfers” instead of “transfer”

Line 166 – “state-managed” (insert dash)

Lines 170-174 – The new text needs a slight rewrite, something like: “In Sri Lanka, forests entered into the CF program were previously state-managed forests. Under state management, the majority of natural forests were protected, and forestry operations such as extraction (e.g. timber) were restricted. However, boundary demarcation and fire-belt establishment were conducted by FD using forest laborers. Furthermore, when the CF program was implemented in forest blocks, the adjacent state-managed forest blocks remained subject to the same FD management activities as before the CF program.” In addition, I think this new text might fit better around line 202.

Line 185 – “impacts” instead of “impact”

Line 204 – “lands” instead of “land”

Line 214 – comma after “Climate”

Line 219 – insert “a” before “low”

Line 252 – rewrite: “…the highest numbers of people who were below the poverty line were…”

Line 286 – delete comma after “sites”

Line 352 – rewrite: “…encroachment plots (Table 2) is a count of the number of locations that violate…”

Table 5 – “Lopped” instead of “Loped”; capitalize the “Before” and “After” columns for the CMBs

Line 497 – “CMBs” instead of “CMB”

Line 534 – delete comma after “income”

Line 559 – “state-managed” (insert dash)

Line 560 – rewrite: “…in seedling and sapling density…”

Line 568 – “state-managed” (insert dash)

Line 584 – “results” instead of “result”

Line 638 – “community-based” (insert dash)

Line 651 – delete comma after “species”

---

## [Author Response · Author response to Decision Letter 2]

31 Aug 2020

Academic Editor Comment 

Comment 1

In lines 593-595 you state that forest boundaries were clearly marked and a fire-belt constructed in the CMBs as a result of the CF program. However, in newly added text (lines 170-174) you indicate that the FD did those things, presumably once the blocks were selected for the CF program. It’s important to be clear about that. In lines 601-605, you assert that the CF program reduced the occurrence of forest fires in both CMBs and SMBs. This conflicts somewhat with Table 5 – yes, the DID coefficient for fire is negative and significant, but the values of the fire presence variable for the SMBs are the same before and after (0.23). So, did the CF program really make a difference in terms of fire in the SMBs? (On the other hand, it seems pretty conclusive for the CMBs.)

Response 1

We are grateful for this comment. As mentioned in the manuscript FD established a fire belt in selected forest areas based on the previous recodes of fire occurrence however, due to high labor cost and limited resources FD unable to maintain the fire belt throughout the year. Once the fire belt opens by FD it will reopen in next year. This is not effective in controlling fire, because most of the grasses (cause for fire spreading) are growing fast and cover the fire belt. Due to this reason, FD encourages CF members to established and maintain a fire belt around CF sites. The current version of the manuscript modified by clarifying these issues. (Lines 202-205)

Furthermore, as indicated by our results, the occurrence of fire in both CMBs and SMBs reduced due to the implementation of the CF program. However, if the fire origin from inside the state forest or another adjacent settlement other than the CF area, those fire cannot be controlled by the fire belt established by the FD. We modified the current version of the manuscript by indicating these reasons (lines 583-584).

Additional Editor Comments

Comment 1

Line 78 – delete comma after “program”

Response

We are grateful to the Additional Editors’ beneficial comments. The current version of the manuscript has been modified by deleting the comma after “program” (line 78).

Comment 2 

Line 79 – change “subsistent” to “subsistence”

Response2

We are grateful to this comment. The current version of the manuscript has been modified by changing “subsistent” to “subsistence” (line 79).

Comment 3

Line 85 – insert comma after “sacred”

Response3

We are grateful to this comment. The current version of the manuscript has been modified by inserting a comma after “sacred” (line 85).

Comment 4

Line 102 – delete “very”

Response 4

We are grateful to this comment. The current version of the manuscript has been modified by deleting “very” (line 102).

Comment 5

Line 150 – spell out non-timber forest products at this first use

Response5

Thank you for the helpful comment. The current version of the manuscript has been modified by spelling out non-timber forest products (line 150).

Comment 6

Line 163 – “transfers” instead of “transfer”

Response 6

Thank you for the helpful comment. The current version of the manuscript has been modified by using “transfers” instead of “transfer” (line 163).

Comment 7

Line 166 – “state-managed” (insert dash)

Response 7

Thank you for the helpful comment. The current version of the manuscript has been modified by inserting dash in “state-managed” (line 166).

Comment 8

Lines 170-174 – The new text needs a slight rewrite, something like: “In Sri Lanka, forests entered into the CF program were previously state-managed forests. Under state management, the majority of natural forests were protected, and forestry operations such as extraction (e.g. timber) were restricted. However, boundary demarcation and fire-belt establishment were conducted by FD using forest laborers. Furthermore, when the CF program was implemented in forest blocks, the adjacent state-managed forest blocks remained subject to the same FD management activities as before the CF program.” In addition, I think this new text might fit better around line 202.

Response 8

Thank you for the helpful comment. The current version of the manuscript has been modified by rewriting the line and we transferred those lines into 202 in the old manuscript (lines 197-202)

Comment 9

Line 185 – “impacts” instead of “impact”

Response 9

We are grateful to this comment. The current version of the manuscript has been modified by inserting “impacts” instead of “impact” (line 180).

Comment 10

Line 204 – “lands” instead of “land”

Response 10

We are grateful to this comment. The current version of the manuscript has been modified by inserting “lands” instead of “land” (line 207).

Comment 11

Line 214 – comma after “Climate”

Response 11

We are grateful to this comment. The current version of the manuscript has been modified by inserting a comma after “Climate” (line 218).

Comment 12

Line 219 – insert “a” before “low”

Response 12

We are grateful to this comment. The current version of the manuscript has been modified by inserting “a” before “low” (line 223)

Comment 13

Line 252 – rewrite: “…the highest numbers of people who were below the poverty line were…”

Response 13

Thank you for the helpful comment. The current version of the manuscript has been modified by rewriting the line (line 240).

Comment 14 

Line 286 – delete comma after “sites”

Response 14

We are grateful to this comment. The current version of the manuscript has been modified by deleting comma after “sites” (line 270).

Comment 15

Line 352 – rewrite: “…encroachment plots (Table 2) is a count of the number of locations that violate…”

Response 15

Thank you for the helpful comment. The current version of the manuscript has been modified by rewriting the line (line 335).

Comment 16

Table 5 – “Lopped” instead of “Loped”; capitalize the “Before” and “After” columns for the CMBs

Response 16

Thank you for the helpful comment. The current version of the manuscript has been modified by correcting Lopped” instead of “Loped”; capitalizing the “Before” and “After” columns for the CMBs (Table 5).

Comment 17

Line 497 – “CMBs” instead of “CMB”

Response 17

We are grateful to this comment. The current version of the manuscript has been modified by inserting “CMBs” instead of “CMB” (line 475).

Comment 18

Line 534 – delete comma after “income”

Response 18

We are grateful to this comment. The current version of the manuscript has been modified by deleting comma after “income” (line 512).

Comment 19

Line 559 – “state-managed” (insert dash)

Response 19

Thank you for the helpful comment. The current version of the manuscript has been modified by inserting dash in “state-managed” (line 537).

Comment 20

Line 560 – rewrite: “…in seedling and sapling density…”

Response 20

Thank you for the helpful comment. The current version of the manuscript has been modified by rewriting the line (line 538).

Comment 21

Line 568 – “state-managed” (insert dash)

Response 21

Thank you for the helpful comment. The current version of the manuscript has been modified by inserting dash in “state-managed” (line 546). 

Comment 22

Line 584 – “results” instead of “result”

Response 22

Thank you for the helpful comment. The current version of the manuscript has been modified by inserting “results” instead of “result” (line 562).

Comment 23

Line 638 – “community-based” (insert dash)

Response 23

Thank you for the helpful comment. The current version of the manuscript has been modified by inserting dash in “community-based” (line 617).

Comment 24

Line 651 – delete comma after “species”

Response 24

Thank you for the helpful comment. The current version of the manuscript has been modified by deleting comma after “species” (line 630).

---

## [Editor Report · Decision Letter 3]

7 Sep 2020

Impacts of community forestry on forest condition: Evidence from Sri Lanka's intermediate zone

PONE-D-20-13177R3

Dear Dr. Xie,

We’re pleased to inform you that your manuscript has been judged scientifically suitable for publication and will be formally accepted for publication once it meets all outstanding technical requirements.

Kind regards,

Frank H. Koch, PhD

Academic Editor

PLOS ONE

Additional Editor Comments (optional):

Thank you for your attention to all of my comments and those of the reviewers on previous versions of the manuscript. I believe the manuscript is now suitable for publication, although there are a couple of edits I would like you to make for the final proof:

Lines 197-208 - rewrite: "In Sri Lanka, forests entered into the CF program were previously state-managed forests. Under state management, the majority of natural forests were protected, and forestry operations such as extraction (e.g. timber) were restricted. In addition, boundary demarcation and fire-belt establishment were conducted by FD using forest laborers. However, due to the high labor cost and limited human and other resources, FD is unable to maintain fire belts throughout the year. To overcome these issues, FD encourages CF members to maintain the fire belts in their CF blocks. Notably, forest blocks which were handed over to the communities were mostly degraded lands and were closer, on average, to the communities than the state-managed blocks. The FD aims to improve these degraded lands through community participation. It should also be noted that when the CF program was implemented in forest blocks, the adjacent state-managed forest blocks remained subject to the same FD management activities as before the CF program." (I think this is a clearer sentence order.)

Lines 610-611 - slight rewrite: "However, a fire belt maintained by CF is ineffective against fires that start inside the state forest or spread from other adjacent settlements away from CF sites."
---

## [Editor Report · Acceptance letter]

17 Sep 2020

PONE-D-20-13177R3 

Impacts of community forestry on forest condition: Evidence from Sri Lanka’s intermediate zone 

Dear Dr. Xie:

I'm pleased to inform you that your manuscript has been deemed suitable for publication in PLOS ONE. Congratulations! Your manuscript is now with our production department. 

Kind regards, 

on behalf of

Dr. Frank H. Koch 

Academic Editor

PLOS ONE